# MetaDCSeg: Robust Medical Image Segmentation via Meta Dynamic Center Weighting

## Abstract

Medical image segmentation is crucial for clinical applications, but it is frequently disrupted by noisy annotations and ambiguous anatomical boundaries, which lead to instability in model training. Existing methods typically rely on global noise assumptions or confidence-based sample selection, which inadequately mitigate the performance degradation caused by annotation noise, especially in challenging boundary regions. To address this issue, we propose MetaDCSeg, a robust framework that dynamically learns optimal pixel-wise weights to suppress the influence of noisy ground-truth labels while preserving reliable annotations. By explicitly modeling boundary uncertainty through a Dynamic Center Distance (DCD) mechanism, our approach utilizes weighted feature distances for foreground, background, and boundary centers, directing the model's attention toward hard-to-segment pixels near ambiguous boundaries. This strategy enables more precise handling of structural boundaries, which are often overlooked by existing methods, and significantly enhances segmentation performance. Extensive experiments across four benchmark datasets with varying noise levels demonstrate that MetaDCSeg consistently outperforms existing state-of-the-art methods.

## 1 Introduction

Medical image segmentation is indispensable for modern healthcare, serving as the foundation for accurate disease diagnosis, treatment planning, and clinical decision-making (Yang et al., 2023; Bai et al., 2023; He et al., 2023). The development of deep learning architectures such as U-Net (Ronneberger et al., 2015), U-Net++ (Zhou et al., 2018), and nnU-Net (Isensee et al., 2021), coupled with the availability of comprehensive datasets including ISIC2018 (Tschandl et al., 2018), BraTS (Menze et al., 2015), LiTS (Bilic et al., 2019), and ACDC (Zhuang et al., 2021), has significantly advanced medical image segmentation. However, achieving precise segmenta-

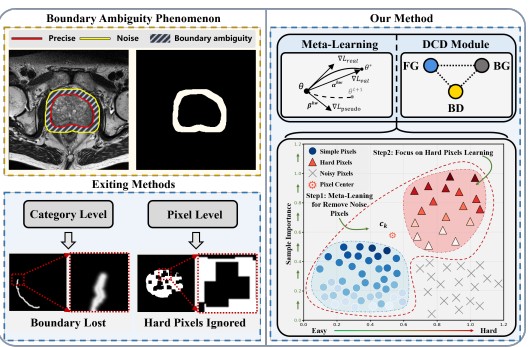

Figure 1: Boundary challenges in medical image segmentation and our framework.

tion remains fundamentally challenging due to the prevalence of noisy labels in medical annotations. In particular, the prevalent crowdsourcing annotation paradigm introduces significant quality concerns. Different annotators exhibit varying levels of expertise and subjective interpretation, leading to inconsistent boundary delineation and annotation errors. Moreover, inherent imaging limitations—including variable illumination conditions (Abhishek et al., 2020), heterogeneous scanning protocols (Gao et al., 2025), device-specific artifacts, and poor tissue contrast (Cossio, 2023; Shorten & Khoshgoftaar, 2019)—further exacerbate boundary ambiguity. Consequently, these compounding factors create substantial uncertainty in distinguishing pathological from normal tissues, resulting in noisy labels that severely compromise model training and clinical reliability.

Recently, noise-robust learning has garnered increasing attention in medical image segmentation, with existing approaches broadly categorized into category-level noise modeling and pixel-level denoising strategies. The first category typically employs noise transition matrices (Yang & Farsiu,

2023) and robust loss (Liang et al., 2023) functions based on predefined assumptions about label noise distribution. However, due to the randomness and diversity of real-world noise, the predefined assumptions these methods rely on limit their ability to effectively handle noise, making them incapable of addressing annotation noise in real-world medical segmentation. The second category focuses on pixel-level denoising using confidence-based sample selection (Liu et al., 2024; Nam et al., 2024), but previous dynamic threshold strategies ignore the differences in segmentation difficulty between classes. In medical images, large structures exhibit well-defined contours and are relatively easy to segment, while smaller structures such as lesion boundaries present ambiguous and morphologically unstable characteristics. This disparity causes models to prioritize easily distinguishable samples while neglecting challenging regions, leading to class imbalance and under-representation of boundary areas. Although recent work Lei et al. (2025) has introduced class-imbalance-aware dynamic thresholds, such approaches fall short in addressing label noise and boundary ambiguity in medical segmentation tasks.

To address these challenges, we propose MetaDCSeg, a novel framework that tackles both label noise and boundary ambiguity through a synergistic two-stage approach. In the first stage, we leverage a meta-learning paradigm to perform pixel-wise reweighting, where cross-entropy losses between ground truth and pseudo labels are computed for each pixel, and meta-learning is applied to learn optimal pixel-wise weights that effectively suppress the influence of noisy annotations. In the second stage, to counteract the model's tendency to focus on easily segmentable regions under noisy supervision, we introduce a dynamic feature distance-guided module that drives the network to pay more attention to informative, hard-to-segment pixels, thereby improving generalization and robustness under noisy boundary annotations. Our experiments demonstrate that MetaDCSeg achieves consistent and significant performance improvements across four real-world medical segmentation benchmarks—MSD Heart, MSD Brain, PROMISE12, and Kvasir-SEG—under varying noise ratios (20%, 40%, and 60%), validating its effectiveness in handling both label noise and boundary uncertainty in medical image segmentation.

The main contributions are summarized as follows:

- We propose MetaDCSeg, a novel framework that seamlessly integrates pixel-wise meta-learning with dynamic center weighting to address both label noise and boundary ambiguity in medical image segmentation, enabling robust learning under noisy supervision.

- We introduce a Dynamic Center Distance (DCD) mechanism that quantifies boundary uncertainty through weighted feature distances to foreground, background, and boundary centers, combined with a meta-learning paradigm that dynamically learns pixel-wise weights.

- Extensive experiments across four benchmark datasets under varying noise ratios demonstrate MetaDCSeg achieves significant performance improvements over state-of-the-art methods.

## 2 RELATED WORKS

### 2.1 MEDICAL IMAGE SEGMENTATION

Medical image segmentation has experienced significant advancements through deep learning, particularly building upon the foundational U-Net architecture with continuous refinements in encoder-decoder designs (Rayed et al., 2024; Wang et al., 2022b; Qureshi et al., 2023). Recent progress encompasses three major directions: enhanced context modeling through Transformer-based architectures such as Swin SMT and FE-SwinUper that leverage soft mixture-of-experts and multi-scale feature integration (Płotka et al., 2024; Zhang et al., 2025b); the emerging "large model + prompt" paradigm exemplified by ProMISe and Med-SAM Adapter that adapt pretrained foundation models for 3D and multimodal scenarios (Li et al., 2023; Wu et al., 2025); and the adoption of diffusion models including LDSeg, cDAL, and DiffuSeg that demonstrate superior performance in high-accuracy, weakly supervised, and cross-domain segmentation tasks (Zaman et al., 2025; Hejrati et al., 2025; Zhang et al., 2025a). However, medical images inherently present difficult boundaries where foreground objects exhibit fuzzy and discontinuous edges due to low contrast, imaging artifacts, and annotation uncertainties, making boundary regions particularly challenging for segmentation models to accurately delineate.

## 2.2 LEARNING WITH NOISY LABELS

Learning with noisy labels poses a fundamental challenge for existing deep learning models, particularly with weak supervision or non-expert annotations. Two primary approaches have emerged: model-based methods estimate noise transition matrices to recover optimal classifiers but struggle with heavy noise or numerous classes (Cheng et al., 2025; Wang et al., 2022a); while model-free methods employ noisy sample detection (Zhang et al., 2020; Jiang et al., 2025) and pseudo-label refinement (Li et al., 2020) to directly mitigate noise impact. However, both paradigms exhibit limitations in high-noise or data-scarce scenarios. Consequently, recent efforts have explored unsupervised contrastive learning (Karim et al., 2022; Zheltonozhskii et al., 2022; Sheng et al., 2024) to learn robust representations without explicit label correction. Nevertheless, achieving scalable and robust representation learning under noisy supervision remains challenging due to the complex interplay between label noise and model generalization.

## 2.3 META LEARNING FOR LABEL CORRECTION

Meta-learning has emerged as a powerful paradigm for enhancing robustness against noisy labels, leveraging small clean validation sets through sophisticated bi-level optimization and instance reweighting strategies (Li et al., 2019; Vettoruzzo et al., 2024; Wu et al., 2021). Key developments in this rapidly evolving field include: WarpGrad (Pham et al., 2019) pioneering bilevel optimization for adaptive sample reweighting; L2B (Zhang et al., 2021) introducing pixel-wise confidence estimation for fine-grained noise handling; Jo-Seg (Guo et al., 2022) innovatively unifying pseudo-label denoising with sample reweighting in a joint framework; and MGL (Liu et al., 2023) exploiting cross-image consistency for enhanced robustness. Recent advances—CMW-Net (Shu et al., 2023) for adaptive weighting under diverse data biases and DMLP (Tu et al., 2023) combining self-supervised learning with meta-learning approaches—further demonstrate meta-learning's remarkable effectiveness in treating label correction as an iterative meta-process (Wu et al., 2021; Zheng et al., 2021; Wang et al., 2025) or learning robust loss functions (Luo et al., 2024; Chen et al., 2025).

# 3 METHODOLOGY

## 3.1 PROBLEM FORMULATION AND OVERVIEW

Given a medical image segmentation dataset with potentially noisy annotations, we denote the training set as $\mathcal{D}_{\text{train}} = \{(\mathbf{x}_i, \mathbf{y}_i)\}_{i=1}^{N}$, where $\mathbf{x}_i \in \mathbb{R}^{H \times W \times C}$ represents the $i$-th input image and $\mathbf{y}_i \in \{0, 1, ..., L-1\}^{H \times W}$ denotes the corresponding pixel-wise labels that may contain annotation noise. Following the meta-learning paradigm, we assume access to a small clean validation set $\mathcal{D}_{\text{val}} = \{(\mathbf{x}_i^v, \mathbf{y}_i^v)\}_{i=1}^{M}$ with $M \ll N$, where $H$, $W$, and $C$ denote the image height, width, and number of channels respectively, and $L$ is the number of segmentation classes. We denote $\mathbf{M}_i \in \{0, 1\}^{H \times W}$ and $\hat{\mathbf{M}}_i \in [0, 1]^{H \times W}$ as the ground truth and predicted masks for image $i$.

## 3.2 META-LEARNING FOR PIXEL-WISE NOISE SUPPRESSION

Medical segmentation faces spatially heterogeneous label noise—annotations are reliable in clear anatomical regions but deteriorate at ambiguous boundaries. To address this, we adopt a pixel-wise meta-learning framework, which dynamically assigns reliability weights to each pixel, enabling adaptive trust in different supervision sources across the image.

### 3.2.1 PIXEL-WISE BOOTSTRAPPING LOSS

For each pixel $(h, w)$ in image $i$, we define the meta-weighted loss as:

$$\mathcal{L}_i^{hw}(\theta) = \alpha_i^{hw} \cdot \mathcal{L}_{\text{CE}}(p_i^{hw}, y_i^{\text{real},hw}) + \beta_i^{hw} \cdot \mathcal{L}_{\text{CE}}(p_i^{hw}, y_i^{\text{pseudo},hw}), \tag{1}$$

where $p_i^{hw} = F(\mathbf{x}_i; \theta)^{hw}$ denotes the predicted probability at pixel $(h, w)$, $y_i^{\text{real},hw}$ is the observed (potentially noisy) label, $y_i^{\text{pseudo},hw} = \arg\max_{c \in \{0,...,L-1\}} p_i^{hw}[c]$ is the pseudo-label, and $\mathcal{L}_{\text{CE}}$ is

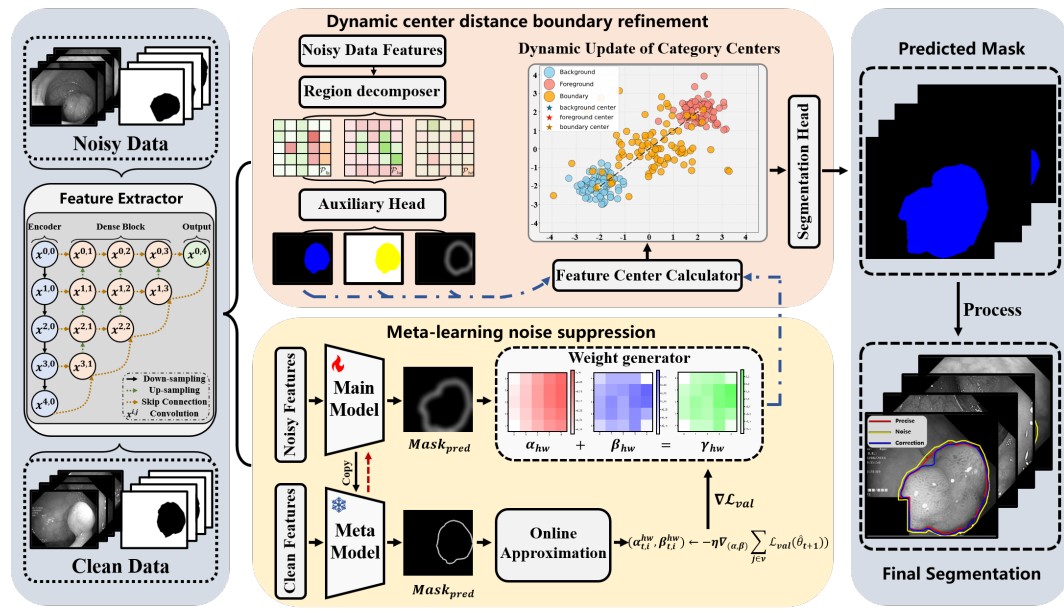

Figure 2: The proposed MetaDCSeg architecture with pixel-wise meta-learning and dynamic boundary refinement modules.

the cross-entropy loss. The meta-learned weights $\alpha_i^{hw}$ and $\beta_i^{hw}$ balance between exploiting ground-truth annotations and leveraging self-supervision: $\alpha_i^{hw}$ dominates in reliable regions to utilize expert knowledge, while $\beta_i^{hw}$ increases at uncertain boundaries to filter annotation noise through model predictions.

### 3.2.2 WEIGHT INITIALIZATION STRATEGY

The meta-learned weights $\alpha_i^{hw}$ and $\beta_i^{hw}$ are initialized based on the model's initial confidence:

$$\begin{aligned}
\alpha_{0,i}^{hw} &= 1.0, \quad \forall(h,w,i), \\
\beta_{0,i}^{hw} &= 0.0, \quad \forall(h,w,i),
\end{aligned} \tag{2}$$

where the subscript $0$ denotes the initial iteration. This initialization assumes initial trust in the provided annotations while allowing the model to gradually learn to incorporate pseudo-labels as training progresses.

### 3.2.3 META-LEARNING OBJECTIVE

To achieve precise pixel-wise noise suppression, our method assigns adaptively learned weights to each pixel. These weights are refined through a bi-level optimization framework that dynamically adjusts $\alpha_i^{hw}$ and $\beta_i^{hw}$ for each pixel based on the validation set.

We denote the weight matrices as $\alpha = \{\alpha_i^{hw} \mid \forall i, h, w\}$ and $\beta = \{\beta_i^{hw} \mid \forall i, h, w\}$, where each pair of weights satisfies the complementary constraint $\alpha_i^{hw} + \beta_i^{hw} = 1$ with $\alpha_i^{hw}, \beta_i^{hw} \in [0,1]$.

The inner optimization learns the model parameters given the current weights:

$$\begin{aligned}
\theta^*(\alpha, \beta) = \arg\min_\theta \sum_{i=1}^{N} \sum_{h,w} [&\alpha_i^{hw} \mathcal{L}_{\text{CE}}(p_i^{hw}, y_i^{\text{real},hw}) \\
&+ \beta_i^{hw} \mathcal{L}_{\text{CE}}(p_i^{hw}, y_i^{\text{pseudo},hw})],
\end{aligned} \tag{3}$$

where $\theta^*(\alpha, \beta)$ represents the model parameters optimized with fixed weight matrices $\alpha$ and $\beta$ for all pixels in the training set.

The outer optimization determines the optimal weights by minimizing the validation loss:

$$\alpha^*, \beta^* = \arg\min_{\alpha,\beta} \mathbb{E}_{(\mathbf{x}^v,\mathbf{y}^v)\sim\mathcal{D}_{\text{val}}} \left[\mathcal{L}_{\text{CE}}(F(\mathbf{x}^v; \theta^*(\alpha,\beta)), \mathbf{y}^v)\right] \tag{4}$$

Note that in Eq. 4, although $\alpha$ and $\beta$ do not appear explicitly in the loss function, they influence the validation loss through the optimized model parameters $\theta^*(\alpha,\beta)$. This bi-level optimization framework prevents the trivial solution ($\alpha = 0, \beta = 0$) through the complementary constraint, ensuring that each pixel receives a meaningful weighted combination of real and pseudo-label supervision.

Through this dual optimization in Eq. 3 and Eq. 4, we can dynamically learn both the model parameters $\theta$ and the pixel-wise weights for real labels $\alpha$ and pseudo-labels $\beta$. The meta-learning objective automatically identifies and reduces the contribution of noisy pixels while emphasizing clean ones, as validated on the clean validation set.

Meanwhile, the online approximation of this bi-level optimization follows the method of Zhou et al. (2024) which is provided in Appendix.A.3.

### 3.3 DYNAMIC CENTER DISTANCE BOUNDARY REFINEMENT

Building upon the meta-learned representations, we introduce a dynamic center weighting mechanism that exploits a key insight: ambiguous boundary pixels exhibit transitional feature characteristics between foreground and background regions in the feature space. The boundary center in Eq. 10 serves as a reference point for measuring these feature transition patterns, capturing the intermediate state where pixels are equidistant from both semantic regions. By measuring weighted distances to these centers, we quantify boundary uncertainty and direct model attention to challenging regions, while meta-learned reliability weights ensure robust center estimation.

#### 3.3.1 FEATURE EXTRACTION

We extract feature representations from the penultimate layer of the decoder in network $F$, which provides a balance between high-level semantic information and spatial resolution. Specifically, for each pixel $(h, w)$ in image $i$, the feature vector is:

$$\mathbf{h}_i^{hw} = F_{\text{feat}}(\mathbf{x}_i; \theta)^{hw} \in \mathbb{R}^D, \tag{5}$$

where $F_{\text{feat}}$ denotes the feature extraction operation up to the penultimate decoder layer, and $D$ is the feature dimension.

#### 3.3.2 UNCERTAINTY-AWARE REGION DECOMPOSITION

We decompose the image into three distinct regions: foreground $\mathcal{P}_{\text{fg}}$ (high-confidence predictions), background $\mathcal{P}_{\text{bg}}$ (low-confidence predictions), and boundary $\mathcal{P}_{\text{bd}}$ (intermediate confidence plus detected edges):

$$\mathcal{P}_{\text{fg},i} = \{(h,w)|\hat{M}_i^{hw} > \tau\}, \tag{6}$$

$$\mathcal{P}_{\text{bg},i} = \{(h,w)|\hat{M}_i^{hw} < 1-\tau\}, \tag{7}$$

$$\mathcal{P}_{\text{bd},i} = \{(h,w)|1-\tau \leq \hat{M}_i^{hw} \leq \tau\} \cup \mathcal{P}_{\text{edge},i}, \tag{8}$$

where $\hat{M}_i^{hw}$ is the predicted foreground probability at pixel $(h, w)$ in image $i$, $\tau \in (0.5, 1)$ is a confidence threshold, and $\mathcal{P}_{\text{edge},i}$ contains edge pixels detected using PiDiNet (Su et al., 2021).

#### 3.3.3 META-WEIGHTED DYNAMIC CENTERS

The uncertainty weight serves as a pixel-wise reliability indicator:

$$\gamma_i^{hw} = \tilde{\alpha}_i^{hw} + \tilde{\beta}_i^{hw}, \tag{9}$$

where $\tilde{\alpha}_i^{hw}$ and $\tilde{\beta}_i^{hw}$ are the rectified meta-learned weights from online approximation, representing the reliability of annotation and model prediction respectively.

The uncertainty weight $\gamma_i^{hw}$ provides a pixel-wise reliability indicator that naturally adapts to medical image characteristics. As illustrated in Fig. 3, high $\gamma_i^{hw}$ values (visualized in yellow) identify reliable pixels corresponding to regions far from ambiguous boundaries, while low values (shown in red to black) indicate uncertain regions near the boundary where annotation ambiguity is highest.

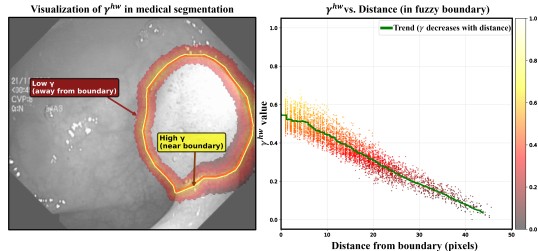

Figure 3: Spatial distribution and distance-dependency of uncertainty weight.

We compute weighted feature centers for each region:

$$\mathbf{c}_{k,i} = \frac{\sum_{(h,w)\in\mathcal{P}_{k,i}} \gamma_i^{hw} \cdot \mathbf{h}_i^{hw}}{\sum_{(h,w)\in\mathcal{P}_{k,i}} \gamma_i^{hw}}, \quad k \in \{\text{fg}, \text{bg}, \text{bd}\}, \tag{10}$$

where $\mathbf{h}_i^{hw} \in \mathbb{R}^D$ is the $D$-dimensional feature representation at pixel $(h,w)$ in image $i$.

### 3.3.4 ADAPTIVE BOUNDARY WEIGHTING

For pixels in the boundary region, we compute a composite distance metric:

$$\text{DCD}_i^{hw} = \frac{\|\mathbf{h}_i^{hw} - \mathbf{c}_{\text{fg},i}\|_2 \cdot \|\mathbf{h}_i^{hw} - \mathbf{c}_{\text{bg},i}\|_2}{\|\mathbf{h}_i^{hw} - \mathbf{c}_{\text{bd},i}\|_2 + \epsilon}, \tag{11}$$

where $\|\cdot\|_2$ denotes the Euclidean norm and $\epsilon = 10^{-8}$ is a small constant for numerical stability.

This metric captures the relative position of boundary pixels with respect to all three centers. The normalized importance weight is:

$$w_i^{hw} = \frac{\exp(\text{DCD}_i^{hw}/\tau_{\text{dcd}})}{\sum_{(h',w')\in\mathcal{P}_{\text{bd},i}} \exp(\text{DCD}_i^{h'w'}/\tau_{\text{dcd}})}, \tag{12}$$

where $\tau_{\text{dcd}} > 0$ is a temperature parameter controlling the sharpness of the weight distribution.

## 3.4 UNIFIED TRAINING OBJECTIVE

The final training objective seamlessly integrates both stages:

$$\mathcal{L}_{\text{total}} = \sum_{i=1}^{N} \left[ \sum_{h,w} \mathcal{L}_i^{hw}(\theta) + \lambda_1 \sum_{(h,w)\in\mathcal{P}_{\text{bd},i}} w_i^{hw} \cdot \mathcal{L}_i^{hw}(\theta) \right]$$
$$+ \lambda_2 \mathcal{L}_{\text{Dice}}(\mathbf{M}_i, \hat{\mathbf{M}}_i), \tag{13}$$

where $\mathcal{L}_i^{hw}(\theta)$ is the meta-weighted pixel-wise loss (Eq. 1), $\lambda_1$, $\lambda_2$ are hyperparameters for boundary refinement and global consistency balance, and $\mathcal{L}_{\text{Dice}}$ ensures global structural consistency.

## 4 EXPERIMENTS

### 4.1 EXPERIMENTAL SETTINGS

**Datasets.** We evaluate MetaDCSeg on four benchmark medical imaging datasets:

- **MSD Heart** (Simpson et al., 2019): A cardiac segmentation subset from the Multi-Atlas Labeling Beyond the Cranial Vault (MSD) challenge, comprising 20 3D CT scans.
- **MSD Brain** (Simpson et al., 2019): A brain tumor segmentation subset containing 484 multi-modal MRI volumes with T1-weighted sequences. The dataset is divided into 387 training volumes and 49 test volumes.
- **PROMISE12** (Litjens et al., 2014): A prostate MRI segmentation dataset featuring 50 T2-weighted 3D scans with expert manual annotations. We employ 40 scans for training and reserve 10 for testing purposes.

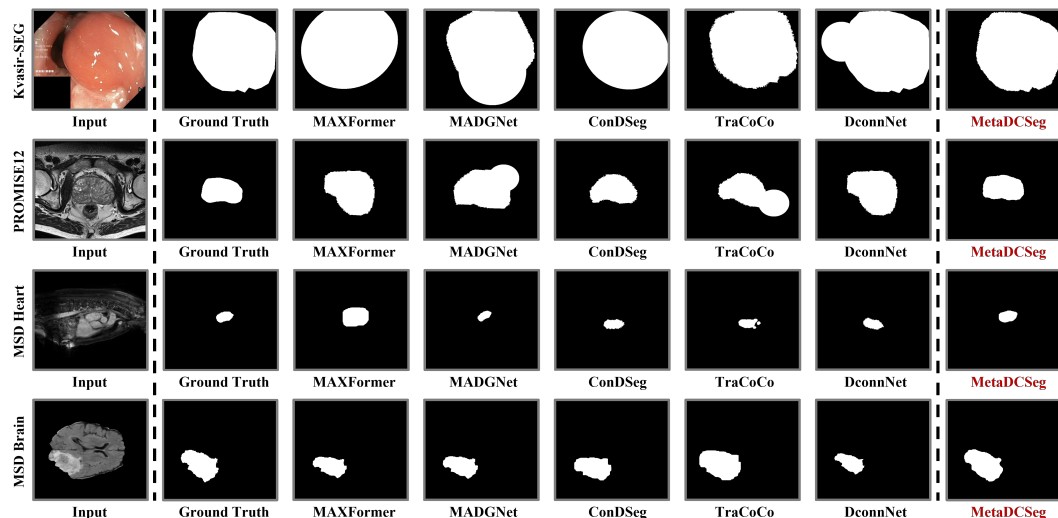

Figure 4: Visual comparison of segmentation methods on Kvasir-SEG, PROMISE12, MSD Heart, and MSD Brain datasets.

Table 1: Performance comparison of the MSD Heart dataset at different noise levels. Results marked with (*) indicate experiments conducted by the authors.

| Noise Level | 0% | | | 20% | | | 40% | | | 60% | | |
|---|---|---|---|---|---|---|---|---|---|---|---|---|
| Method | mIoU ↑ | DSC ↑ | HD ↓ | mIoU ↑ | DSC ↑ | HD ↓ | mIoU ↑ | DSC ↑ | HD ↓ | mIoU ↑ | DSC ↑ | HD ↓ |
| MAXFormer* | 68.17 | 81.32 | 15.34 | 65.34 | 79.45 | 19.15 | 62.24 | 76.72 | 24.70 | 51.80 | 68.25 | 30.52 |
| MADGNet* | 75.16 | 83.27 | 14.32 | 70.77 | 82.88 | 21.06 | 65.62 | 79.24 | 19.59 | 63.69 | 77.82 | 28.31 |
| ConDSeg* | 67.91 | 76.17 | 22.53 | 62.82 | 73.07 | 30.17 | 58.35 | 69.05 | 29.53 | 63.22 | 73.21 | 27.41 |
| HAMIL* | 53.26 | 69.14 | 17.31 | 50.59 | 62.56 | 19.37 | 65.32 | 75.79 | 16.55 | 53.32 | 64.59 | 23.50 |
| L2B* | 77.92 | 87.17 | 13.23 | 73.56 | 84.76 | 16.17 | 71.60 | 83.45 | 14.45 | 65.89 | 79.44 | 17.84 |
| MetaDCSeg | **79.58** | **89.14** | **5.76** | **77.23** | **86.63** | **8.36** | **75.73** | **84.59** | **12.28** | **71.86** | **81.19** | **15.26** |
| Δ | ↑ **1.66** | ↑ **1.97** | ↓ **7.47** | ↑ **3.67** | ↑ **1.87** | ↓ **7.81** | ↑ **4.13** | ↑ **1.14** | ↓ **2.17** | ↑ **5.97** | ↑ **1.75** | ↓ **2.58** |

- **Kvasir-SEG** (Jha et al., 2020): A polyp segmentation dataset composed of 1,000 high-resolution endoscopic images collected during colonoscopy procedures. Following established protocols, we utilize 800 images for training, 100 for validation, and 100 for testing.

**Evaluation Metrics.** We employ four standard metrics: **mIoU** (mean Intersection over Union), **DSC** (Dice Similarity Coefficient) and **HD** (Hausdorff Distance). Higher mIoU/DSC and lower HD indicate better performance.

**Implementation Details.** MetaDCSeg uses U-Net++ backbone implemented in PyTorch. Training configuration: 100 epochs with SGD (lr=0.005, momentum=0.9, weight decay=$5 \times 10^{-4}$), LambdaLR scheduler ($\lambda = 20/(epoch+20)$), and 10-epoch warm-up. Meta-learning employs single-step gradient updates with 2% clean meta-dataset. Key hyperparameters: momentum $\alpha = 0.9$, GMM threshold $\beta = 3.0$ (linear decay), gradient clipping=0.2, EMA decay=0.99. Data augmentation includes random flipping, rotation ($\pm 10°$), and multi-scale zoom [4-30 pixels]. The noise injection methods for image segmentation tasks follow the approach proposed in Zhou et al. (2024), with detailed implementations provided in Appendix A.2.

### 4.2 COMPARISON WITH THE STATE-OF-THE-ARTS

We evaluate MetaDCSeg against five state-of-the-art methods across four medical segmentation datasets under clean conditions (0% noise) and noisy supervision (20%, 40%, and 60% noise levels). Importantly, as shown in Table 1, MetaDCSeg achieves 79.58% mIoU on clean MSD Heart data, surpassing L2B (Zhou et al., 2024) by 1.66%, demonstrating that our noise-handling com-

Table 2: The performance of the Kvasir-SEG (colon) dataset at different noise levels.

| Noise Level | 0% | | | 20% | | | 40% | | | 60% | | |
|---|---|---|---|---|---|---|---|---|---|---|---|---|
| Method | mIoU ↑ | DSC ↑ | HD ↓ | mIoU ↑ | DSC ↑ | HD ↓ | mIoU ↑ | DSC ↑ | HD ↓ | mIoU ↑ | DSC ↑ | HD ↓ |
| MAXFormer * | 68.42 | 81.25 | 35.18 | 54.65 | 70.67 | 27.87 | 34.99 | 51.28 | 61.05 | 30.55 | 46.80 | 49.21 |
| MADGNet * | 75.63 | 86.14 | 28.92 | 64.29 | 76.90 | 53.16 | 53.30 | 67.56 | 61.09 | 39.60 | 54.07 | 78.38 |
| ConDSeg * | 82.76 | 90.68 | 18.35 | 70.88 | 82.96 | 31.47 | 53.72 | 69.89 | 39.53 | 34.66 | 51.47 | 53.07 |
| HAMIL * | 84.28 | 91.52 | 12.64 | 70.54 | 79.36 | 19.07 | 67.67 | 76.01 | 20.81 | 64.32 | 68.77 | 22.54 |
| L2B * | 91.85 | 95.78 | 6.23 | 88.19 | 93.62 | 10.44 | 81.75 | 88.94 | 18.57 | 72.83 | 82.96 | 31.87 |
| MetaDCSeg | **93.47** | **96.35** | **5.16** | **90.89** | **94.42** | **8.07** | **87.01** | **90.72** | **13.96** | **81.51** | **85.19** | **18.67** |
| Δ | ↑ **1.62** | ↑ **0.57** | ↓ **1.07** | ↑ **2.70** | ↑ **0.80** | ↓ **2.37** | ↑ **5.26** | ↑ **1.78** | ↓ **4.61** | ↑ **8.68** | ↑ **2.23** | ↓ **3.87** |

Table 3: Performance comparison of the MSD brain dataset at different noise levels.

| Noise Level | 0% | | | 20% | | | 40% | | | 60% | | |
|---|---|---|---|---|---|---|---|---|---|---|---|---|
| Method | mIoU ↑ | DSC ↑ | HD ↓ | mIoU ↑ | DSC ↑ | HD ↓ | mIoU ↑ | DSC ↑ | HD ↓ | mIoU ↑ | DSC ↑ | HD ↓ |
| MAXFormer * | 26.84 | 42.31 | 16.25 | 19.62 | 32.81 | 21.30 | 14.45 | 24.56 | 29.78 | 10.75 | 19.42 | 32.03 |
| MADGNet * | 24.76 | 39.68 | 15.83 | 18.57 | 31.23 | 21.71 | 17.93 | 30.41 | 16.28 | 10.45 | 10.45 | 26.64 |
| ConDSeg * | 27.58 | 43.16 | 14.75 | 21.19 | 34.97 | 18.13 | 18.67 | 36.71 | 19.41 | 15.12 | 34.91 | 20.38 |
| HAMIL * | 28.93 | 44.87 | 12.48 | 20.58 | 42.56 | 15.37 | 15.32 | 35.78 | 18.38 | 14.32 | 25.69 | 25.38 |
| L2B * | 69.45 | 81.97 | 14.62 | 62.72 | 74.59 | 16.28 | 58.52 | 69.74 | 19.12 | 53.74 | 70.66 | 21.85 |
| MetaDCSeg | **77.82** | **87.55** | **11.24** | **73.15** | **79.37** | **13.36** | **69.61** | **74.83** | **15.57** | **67.54** | **72.88** | **18.64** |
| Δ | ↑ **8.37** | ↑ **5.58** | ↓ **1.24** | ↑ **10.43** | ↑ **4.78** | ↓ **2.01** | ↑ **11.09** | ↑ **5.09** | ↓ **0.71** | ↑ **13.80** | ↑ **2.22** | ↓ **1.74** |

ponents do not compromise performance on high-quality annotations. This trend continues across datasets: on Kvasir-SEG (Table 2), MetaDCSeg achieves 93.47% mIoU at 0% noise, improving 1.62% over L2B; on MSD Brain (Table 3), we achieve 77.82% mIoU, an 8.37% improvement over ConDSeg (Lei et al., 2025); and on PROMISE12 (Table 4), we reach 87.15% mIoU, surpassing HAMIL (Zhong et al., 2023) by 1.48%. These clean-data results validate that our meta-learning framework introduces minimal overhead when noise is absent. As noise increases, MetaDCSeg's superiority becomes more pronounced: at 60% noise, we maintain 71.86% mIoU on MSD Heart (5.97% above L2B), 81.51% on Kvasir-SEG (8.68% above L2B), and remarkably 67.54% on MSD Brain where MAXFormer (Liang et al., 2023) catastrophically fails at 10.75%. On PROMISE12 at 60% noise, MetaDCSeg achieves 80.49% mIoU, a 13.96% improvement over HAMIL, while MADGNet (Nam et al., 2024) collapses to merely 4.10%. The consistent improvements from 0% to 60% noise demonstrate that MetaDCSeg not only excels under noisy supervision but also maintains competitive or superior performance on clean data, confirming our approach effectively adapts to varying annotation quality without sacrificing baseline accuracy.

## 5 DISCUSSION

### 5.1 MODULE ABLATION

We systematically evaluate three core components under 40% noise conditions.

**Meta-Learning Module.** As shown in Table 5, removing the meta-learning component causes the most significant performance degradation, with mIoU dropping from 75.73% to 71.42%. This confirms its critical role in noise suppression through adaptive weight assignment between noisy and pseudo labels, enabling robust discrimination of reliable annotations in ambiguous regions.

**DCD Weighting.** The absence of dynamic center distance weighting results in a 2.45% mIoU decrease (75.73%→73.28%) and notable boundary metric degradation (HD: 12.28→13.52). This validates that explicit boundary modeling through distance-based weighting effectively addresses anatomical boundary ambiguities inherent in medical images.

**Dice Loss.** While contributing the smallest individual improvement, removing the Dice loss still causes a 1.12% mIoU drop (75.73%→74.61%). This demonstrates that pixel-wise optimization alone cannot maintain coherent anatomical structures, necessitating region-level supervision for robust segmentation.

Table 4: Performance comparison of the PROMISE12 dataset at different noise levels.

| Noise Level | 0% | | | 20% | | | 40% | | | 60% | | |
|---|---|---|---|---|---|---|---|---|---|---|---|---|
| Method | mIoU ↑ | DSC ↑ | HD ↓ | mIoU ↑ | DSC ↑ | HD ↓ | mIoU ↑ | DSC ↑ | HD ↓ | mIoU ↑ | DSC ↑ | HD ↓ |
| MAXFormer * | 64.75 | 78.62 | 86.43 | 37.28 | 54.31 | 103.69 | 40.76 | 57.92 | 102.97 | 56.96 | 72.58 | 94.21 |
| MADGNet * | 8.92 | 16.38 | 18.26 | 6.30 | 9.40 | 25.78 | 5.46 | 8.05 | 36.33 | 4.10 | 6.55 | 44.69 |
| ConDSeg * | 68.34 | 81.15 | 7.82 | 54.92 | 63.98 | 11.23 | 41.83 | 52.55 | 17.18 | 40.31 | 53.08 | 15.10 |
| HAMIL * | 81.67 | 82.34 | 6.15 | 72.46 | 78.74 | 9.26 | 68.26 | 66.24 | 10.12 | 56.53 | 61.29 | 14.46 |
| L2B | 85.23 | 87.08 | 4.19 | 81.37 | 82.01 | 5.54 | 74.51 | 80.83 | 6.68 | 61.72 | 82.01 | 10.17 |
| MetaDCSeg | **86.71** | **87.92** | **4.01** | **83.70** | **83.99** | **4.52** | **77.99** | **83.32** | **5.28** | **75.68** | **82.75** | **8.84** |
| Δ | ↑ 1.48 | ↑ 0.84 | ↓ 0.18 | ↑ 2.33 | ↑ 1.98 | ↓ 1.02 | ↑ 3.48 | ↑ 2.49 | ↓ 1.40 | ↑ 13.96 | ↑ 0.74 | ↓ 1.33 |

Table 5: Ablation study on different modules of the proposed MetaDCSeg. ✓ indicates the module is included.

| Meta | DCD | Dice Loss | mIoU | DSC | HD |
|---|---|---|---|---|---|
| ✓ | ✓ | ✓ | **75.73** | **84.59** | **12.28** |
| | ✓ | ✓ | 71.42 | 82.15 | 14.76 |
| ✓ | | ✓ | 73.28 | 83.36 | 13.52 |
| ✓ | ✓ | | 74.61 | 83.94 | 12.89 |

Table 6: Performance comparison across meta-dataset sizes. CE: performance gain per computational unit; $M_s$: meta-dataset size.

| $M_s$ | mIoU | DSC | HD | Time/Epoch | Memory | CE |
|---|---|---|---|---|---|---|
| 1% | 74.85 | 84.12 | 13.04 | 2.8 min | 7.2 GB | - |
| **2%** | **75.73** | **84.59** | **12.28** | **3.2 min** | **7.8 GB** | **3.70** |
| 5% | 75.91 | 84.68 | 12.15 | 4.5 min | 9.6 GB | 0.93 |
| 10% | 76.02 | 84.74 | 12.08 | 6.8 min | 12.4 GB | 0.37 |

Table 7: Computational complexity and performance analysis on MSD Heart (40% noise)

| Method/Module | FLOPs (G) | Memory (GB) | Train Time (min/epoch) | Inference (ms/img) | mIoU (%) | ΔmIoU |
|---|---|---|---|---|---|---|
| Baseline U-Net++ | 91.65 | 5.80 | 2.10 | 45.14 | 71.42 | – |
| + Meta Module | 132.00 (+44.0%) | 7.80 (+34.5%) | 3.00 (+42.9%) | 45.31 | 74.51 | +3.09 |
| + DCD Module | 97.80 (+6.7%) | 5.80 (+0.0%) | 2.20 (+4.8%) | 45.34 | 72.18 | +0.76 |
| + Dice Loss | 92.11 (+0.5%) | 5.80 (+0.0%) | 2.14 (+1.9%) | 45.19 | 71.91 | +0.49 |
| **MetaDCSeg** | **141.60 (+54.5%)** | **7.80 (+34.5%)** | **3.20 (+52.4%)** | **45.21** | **75.73** | **+4.31** |

## 5.2 IMPACT OF META-LEARNING DATASET SIZE

We evaluate meta-dataset size impact on the MSD Heart dataset with 40% label noise. To quantify the performance-cost trade-off, we define Cost Efficiency (CE) as:

$$CE = \frac{\Delta mIoU}{(\text{Time}_i/\text{Time}_{1\%}) \times (\text{Memory}_i/\text{Memory}_{1\%}) - 1} \tag{14}$$

where $\Delta mIoU$ denotes performance gain and denominator captures resource overhead.

Table 6 shows that 2% configuration achieves optimal CE (3.70). While 10% improves mIoU by 1.17%, it requires 2.4× training time and 1.7× memory. The CE drops 90% from 3.70 (2%) to 0.37 (10%), indicating severe diminishing returns. Therefore, we adopt 2% as default meta-dataset size.

## 5.3 COMPUTATIONAL COMPLEXITY ANALYSIS

Table 7 demonstrates that MetaDCSeg achieves a significant 4.31% mIoU improvement (75.73% vs. 71.42%) with 54.5% FLOPs and 34.5% memory overhead. The meta-learning module contributes the most (3.09% mIoU gain) despite higher computational cost, while the DCD module efficiently adds 0.76% improvement with only 6.7% FLOPs increase. Crucially, inference time remains nearly identical (45.21ms vs. 45.14ms), preserving real-time capability. Despite 52.4% longer training, MetaDCSeg effectively balances accuracy and efficiency for noisy medical data segmentation.

## 6 CONCLUSION

This paper proposed MetaDCSeg, a medical image segmentation framework addressing noisy annotations and boundary ambiguity. The core idea integrates pixel-wise meta-learning with dynamic center distance weighting. MetaDCSeg's effectiveness was demonstrated through experiments on four benchmark datasets, showing it outperforms existing methods under various noise levels. Ablation studies validated each component's importance, revealing meta-learning contributes most to performance gains.

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

## A    APPENDIX

### A.1    REPRODUCIBILITY STATEMENT

Table 8: Customized hyper-parameters per dataset for MetaDCSeg.

| Dataset | Input Size | Train Batch | Meta Batch | Epochs (total/warm) | LR | WD | Meta Step | Meta LR Scale | $\lambda_{bd}$ | $\lambda_{dice}$ | $\tau_{dcd}$ | Boundary Ratio | Grad Thresh |
|---|---|---|---|---|---|---|---|---|---|---|---|---|---|
| Kvasir-SEG | 256×256 | 2 | 2 | 140 / 10 | 0.0060 | 5e-4 | 1 | 0.01 | 0.35 | 0.08 | 1.0 | 0.14 | 0.08 |
| PROMISE12 | 144×144 | 1 | 2 | 120 / 10 | 0.0050 | 5e-4 | 1 | 0.01 | 0.30 | 0.10 | 1.0 | 0.10 | 0.10 |
| MSD Heart | 160×160 | 1 | 2 | 160 / 15 | 0.0040 | 5e-4 | 1 | 0.01 | 0.33 | 0.12 | 1.0 | 0.11 | 0.10 |
| MSD Brain | 160×160 | 1 | 2 | 180 / 15 | 0.0035 | 4e-4 | 1 | 0.01 | 0.28 | 0.14 | 1.0 | 0.12 | 0.12 |

To ensure the reproducibility of our experiments, we provide core training code at **https://anonymous.4open.science/r/Meseg-6F8A/**. All experiments were conducted on 4 NVIDIA A6000 GPUs. The complete hyperparameter configurations used in our experiments are detailed in Table 8. The repository contains all necessary scripts for data preprocessing, model training, and evaluation to facilitate the reproduction of our results.

## A.2 Data Noise Injection Methods

### A.2.1 Overview of Label Corruption Strategy

To evaluate the robustness of our MetaDCSeg framework against annotation noise, we systematically inject controlled label corruption into the ground truth segmentation masks. Our noise injection strategy simulates realistic annotation errors commonly encountered in medical image segmentation, including boundary imprecision, structural deformation, and topological changes.

### A.2.2 Noise Injection Pipeline

Given a clean ground truth segmentation mask $\mathbf{y}_i \in \{0,1\}^{H \times W}$ for image $i$, we generate the corrupted label $\tilde{\mathbf{y}}_i$ through a multi-stage stochastic process:

Geometric Transformation First, we apply a random rotation to simulate annotator variability in orientation perception:

$$\mathbf{y}_i^{(1)} = \mathcal{R}_\theta(\mathbf{y}_i), \quad \theta \sim \mathcal{U}(-20, 20), \tag{15}$$

where $\mathcal{R}_\theta$ denotes the rotation operator and $\mathcal{U}$ represents uniform distribution.

Morphological Perturbation We then apply morphological operations to simulate boundary annotation errors. With probability $p_{morph} = 0.5$, we apply either erosion or dilation:

**Erosion Operation:**

$$\mathbf{y}_i^{(2)} = \mathbf{y}_i^{(1)} \ominus \mathcal{K}_j, \quad j \sim \mathcal{C}(p_{ero}), \tag{16}$$

**Dilation Operation:**

$$\mathbf{y}_i^{(2)} = \mathbf{y}_i^{(1)} \oplus \mathcal{K}_j, \quad j \sim \mathcal{C}(p_{dil}), \tag{17}$$

where $\ominus$ and $\oplus$ denote morphological erosion and dilation respectively, $\mathcal{K}_j$ represents the structuring element (kernel) of size $j$, and $\mathcal{C}(p)$ is a categorical distribution with probability vector $p$.

Structural Replacement With probability $p_{ellipse} = 0.5$, we replace the segmentation with an elliptical approximation to simulate gross annotation errors:

$$\tilde{\mathbf{y}}_i = \begin{cases} \mathcal{E}(\mathbf{y}_i^{(2)}) & \text{if } r < p_{ellipse} \text{ and } \sum_{h,w} \mathbf{y}_{i,hw}^{(2)} > 300 \\ \mathbf{y}_i^{(2)} & \text{otherwise} \end{cases}, \tag{18}$$

where $r \sim \mathcal{U}(0, 1)$ and $\mathcal{E}(\cdot)$ constructs an ellipse with semi-axes and center derived from the bounding box of the input mask:

$$\mathcal{E}(\mathbf{y})_{hw} = \mathbb{1}\left[\frac{(h - y_c)^2}{r_y^2} + \frac{(w - x_c)^2}{r_x^2} \leq 1\right], \tag{19}$$

with center $(x_c, y_c) = \frac{1}{2}(x_{min} + x_{max}, y_{min} + y_{max})$ and semi-axes $r_x = \frac{1}{2}(x_{max} - x_{min})$, $r_y = \frac{1}{2}(y_{max} - y_{min})$.

### A.2.3 Noise Level Calibration

To achieve target corruption rates of 20%, 40%, and 60%, we calibrate the kernel sizes and probability distributions for each noise level:

The probability vectors $p_{ero}$ and $p_{dil}$ define cumulative thresholds for selecting kernel sizes, where larger kernels produce more severe morphological changes.

### A.2.4 Corruption Rate Verification

The actual corruption rate is measured using the Dice coefficient between original and corrupted labels:

$$\text{Corruption Rate} = 1 - \text{DSC}(\mathbf{y}, \tilde{\mathbf{y}}) = 1 - \frac{2|\mathbf{y} \cap \tilde{\mathbf{y}}|}{|\mathbf{y}| + |\tilde{\mathbf{y}}|}, \tag{20}$$

where $|\cdot|$ denotes the number of positive pixels.

Table 9: Noise injection parameters for different corruption levels

| Parameter | 20% Corruption | 40% Corruption |
|---|---|---|
| **Kernel Sizes** | $\{7^2, 14^2, 21^2,$ $28^2, 35^2\}$ | $\{7^2, 11^2, 13^2,$ $17^2, 21^2\}$ |
| $p_{ero}$ | $[0.1, 0.2, 0.5,$ $0.7, 1.0]$ | $[0.4, 0.6, 0.8,$ $0.9, 1.0]$ |
| $p_{dil}$ | $[0.1, 0.2, 0.5,$ $0.7, 1.0]$ | $[0.4, 0.6, 0.8,$ $0.9, 1.0]$ |

## A.3 ONLINE APPROXIMATION

Directly solving the bi-level optimization is computationally expensive. We adopt an online approximation in Zhou et al. (2024) that alternates between model and weight updates at each iteration $t$.

At each iteration, we first compute a temporary parameter update using the current pixel-wise weights:

$$\hat{\theta}_{t+1} = \theta_t - \lambda \nabla_\theta \sum_{i \in \mathcal{B}} \sum_{h,w} \left[ \alpha_{t,i}^{hw} f_i^{hw}(\theta_t) + \beta_{t,i}^{hw} g_i^{hw}(\theta_t) \right], \tag{21}$$

where $\mathcal{B}$ denotes the current mini-batch, $\lambda$ is the learning rate, $f_i^{hw}(\theta) = \mathcal{L}_{\text{CE}}(F(\mathbf{x}_i; \theta)^{hw}, y_i^{\text{real},hw})$ and $g_i^{hw}(\theta) = \mathcal{L}_{\text{CE}}(F(\mathbf{x}_i; \theta)^{hw}, y_i^{\text{pseudo},hw})$.

With these temporarily updated parameters, we then evaluate the model performance on the validation set and use this feedback to update the meta-weights through gradient descent:

$$(\alpha_{t+1,i}^{hw}, \beta_{t+1,i}^{hw}) = (\alpha_{t,i}^{hw}, \beta_{t,i}^{hw}) - \eta \nabla_{\alpha_{t,i}^{hw}/\beta_{t,i}^{hw}} \mathcal{L}_{\text{val}}(\hat{\theta}_{t+1}), \tag{22}$$

where $\mathcal{V}$ denotes the validation mini-batch, $\eta$ is the meta-learning rate, and $\mathcal{L}_{\text{val}}(\hat{\theta}_{t+1}) = \sum_{h,w} \mathcal{L}_{\text{CE}}(F(\mathbf{x}_j^v; \hat{\theta}_{t+1})^{hw}, y_j^{v,hw})$.

Since the gradient-based update may produce negative weights, we apply rectification to ensure non-negativity, followed by normalization to maintain numerical stability:

$$\tilde{\alpha}_{t,i}^{hw} = \max(\alpha_{t+1,i}^{hw}, 0), \quad \tilde{\beta}_{t,i}^{hw} = \max(\beta_{t+1,i}^{hw}, 0), \tag{23}$$

$$\hat{\alpha}_{t,i}^{hw} = \frac{\tilde{\alpha}_{t,i}^{hw}}{Z_i}, \quad \hat{\beta}_{t,i}^{hw} = \frac{\tilde{\beta}_{t,i}^{hw}}{Z_i}, \tag{24}$$

where $Z_i = \sum_{h,w} [\tilde{\alpha}_{t,i}^{hw} + \tilde{\beta}_{t,i}^{hw}]$ is the normalization factor ensuring numerical stability across the batch.

## A.4 THEORETICAL ANALYSIS

We establish theoretical guarantees for MetaDCSeg's convergence and stability properties. Our analysis demonstrates that the proposed framework achieves robust segmentation under noisy supervision through principled optimization.

### A.4.1 CONVERGENCE OF META-LEARNING OPTIMIZATION

**Theorem 1.** Under the assumptions that (i) the loss functions have $\sigma$-bounded gradients: $\|\nabla f_i^{hw}(\theta)\| \leq \sigma$ and $\|\nabla g_i^{hw}(\theta)\| \leq \sigma$ for all $i, h, w$, where

$$f_i^{hw}(\theta) = \mathcal{L}_{CE}(F(\mathbf{x}_i; \theta)^{hw}, y_i^{\text{real},hw})$$
$$g_i^{hw}(\theta) = \mathcal{L}_{CE}(F(\mathbf{x}_i; \theta)^{hw}, y_i^{\text{pseudo},hw}) \tag{25}$$

and (ii) the validation loss $\mathcal{L}_{\text{val}}(\theta)$ is $L$-Lipschitz smooth, the validation loss monotonically decreases with appropriate learning rates.

**Proof.** Starting from the Lipschitz smoothness condition:

$$
\mathcal{L}_{\text{val}}(\theta_{t+1}) \leq \mathcal{L}_{\text{val}}(\theta_t) + \langle \nabla \mathcal{L}_{\text{val}}(\theta_t), \theta_{t+1} - \theta_t \rangle + \frac{L}{2} \|\theta_{t+1} - \theta_t\|^2 \tag{26}
$$

From our update rule, we have:

$$
\theta_{t+1} - \theta_t = -\lambda \nabla_\theta \sum_{i \in \mathcal{B}} \sum_{h,w} \left[ \hat{\alpha}_{t,i}^{hw} f_i^{hw}(\theta_t) + \hat{\beta}_{t,i}^{hw} g_i^{hw}(\theta_t) \right] \tag{27}
$$

where $\hat{\alpha}_{t,i}^{hw}$ and $\hat{\beta}_{t,i}^{hw}$ are the normalized weights after rectification.

The meta-weight update follows from the bi-level optimization:

$$
\alpha_{t+1,i}^{hw} = \alpha_{t,i}^{hw} - \eta \frac{\partial \mathcal{L}_{\text{val}}(\hat{\theta}_{t+1})}{\partial \alpha_{t,i}^{hw}} \tag{28}
$$

By the chain rule:

$$
\frac{\partial \mathcal{L}_{\text{val}}(\hat{\theta}_{t+1})}{\partial \alpha_{t,i}^{hw}} = \nabla_\theta \mathcal{L}_{\text{val}}(\hat{\theta}_{t+1})^T \cdot \frac{\partial \hat{\theta}_{t+1}}{\partial \alpha_{t,i}^{hw}}
$$
$$
= -\lambda \nabla_\theta \mathcal{L}_{\text{val}}(\hat{\theta}_{t+1})^T \nabla_\theta f_i^{hw}(\theta_t) \tag{29}
$$

Therefore:

$$
\alpha_{t+1,i}^{hw} = \alpha_{t,i}^{hw} + \eta \lambda \langle \nabla_\theta \mathcal{L}_{\text{val}}(\hat{\theta}_{t+1}), \nabla_\theta f_i^{hw}(\theta_t) \rangle \tag{30}
$$

This update increases $\alpha_{t,i}^{hw}$ when the gradient alignment is positive, indicating that increasing this pixel's weight reduces validation loss.

Substituting into the smoothness inequality:

$$
\mathcal{L}_{\text{val}}(\theta_{t+1}) \leq \mathcal{L}_{\text{val}}(\theta_t)
$$
$$
- \lambda \left\langle \nabla \mathcal{L}_{\text{val}}(\theta_t), \nabla_\theta \sum_{i \in \mathcal{B}} \sum_{h,w} \left[ \hat{\alpha}_{t,i}^{hw} f_i^{hw}(\theta_t) + \hat{\beta}_{t,i}^{hw} g_i^{hw}(\theta_t) \right] \right\rangle
$$
$$
+ \frac{L\lambda^2}{2} \left\| \nabla_\theta \sum_{i \in \mathcal{B}} \sum_{h,w} \left[ \hat{\alpha}_{t,i}^{hw} f_i^{hw}(\theta_t) + \hat{\beta}_{t,i}^{hw} g_i^{hw}(\theta_t) \right] \right\|^2 \tag{31}
$$

For the first term, using the fact that meta-weights are updated to align with validation gradients:

$$
- \lambda \left\langle \nabla \mathcal{L}_{\text{val}}(\theta_t), \nabla_\theta \sum_{i,h,w} \hat{\alpha}_{t,i}^{hw} f_i^{hw}(\theta_t) \right\rangle
$$
$$
\approx -\frac{\eta \lambda^2}{Z} \sum_{i \in \mathcal{B}} \sum_{h,w} \langle \nabla \mathcal{L}_{\text{val}}(\theta_t), \nabla_\theta f_i^{hw}(\theta_t) \rangle^2 \tag{32}
$$

where $Z = \sum_{j \in \mathcal{B}} \sum_{h',w'} [\tilde{\alpha}_{t,j}^{h'w'} + \tilde{\beta}_{t,j}^{h'w'}]$ is the normalization factor.

For the second term, using the bounded gradient assumption and the fact that normalized weights sum to 1:

$$\frac{L\lambda^2}{2} \left\| \nabla_\theta \sum_{i,h,w} \left[ \hat{\alpha}_{t,i}^{hw} f_i^{hw}(\theta_t) + \hat{\beta}_{t,i}^{hw} g_i^{hw}(\theta_t) \right] \right\|^2$$
$$\leq \frac{L\lambda^2\sigma^2}{2} \tag{33}$$

Combining these results:

$$\mathcal{L}_{\text{val}}(\theta_{t+1}) \leq \mathcal{L}_{\text{val}}(\theta_t)$$
$$- \frac{\eta\lambda^2}{Z} \sum_{i,h,w} \left[ \langle \nabla\mathcal{L}_{\text{val}}, \nabla f_i^{hw} \rangle^2 \right.$$
$$\left. + \langle \nabla\mathcal{L}_{\text{val}}, \nabla g_i^{hw} \rangle^2 \right]$$
$$+ \frac{L\lambda^2\sigma^2}{2} \tag{34}$$

For convergence, we require the negative term to dominate:

$$\frac{\eta\lambda^2}{Z} \sum_{i,h,w} \left[ \langle \nabla\mathcal{L}_{\text{val}}, \nabla f_i^{hw} \rangle^2 + \langle \nabla\mathcal{L}_{\text{val}}, \nabla g_i^{hw} \rangle^2 \right]$$
$$> \frac{L\lambda^2\sigma^2}{2} \tag{35}$$

This is satisfied when $\lambda < \sqrt{\frac{2}{\eta\sigma^2 ML}}$, where $M = |\mathcal{V}|$ is the validation set size, completing the proof.

### A.4.2 STABILITY ANALYSIS OF DYNAMIC CENTER DISTANCE

**Theorem 2.** The Dynamic Center Distance (DCD) mechanism maintains numerical stability with bounded values and gradients under the following conditions.

**Proof.** For any boundary pixel $(h, w)$, the DCD is computed as:

$$\text{DCD}_i^{hw} = \frac{\|h_i^{hw} - c_{\text{fg},i}\|_2 \cdot \|h_i^{hw} - c_{\text{bg},i}\|_2}{\|h_i^{hw} - c_{\text{bd},i}\|_2 + \epsilon} \tag{36}$$

Let $R = \max_{i,h,w} \|h_i^{hw}\|_2$ be the maximum feature norm. Using the triangle inequality:

$$\|h_i^{hw} - c_{k,i}\|_2 \leq \|h_i^{hw}\|_2 + \|c_{k,i}\|_2$$
$$\leq R + \|c_{k,i}\|_2 \tag{37}$$

Since centers are weighted averages of features:

$$\|c_{k,i}\|_2 = \left\| \frac{\sum_{(h',w')\in\mathcal{P}_{k,i}} \gamma_i^{h'w'} h_i^{h'w'}}{\sum_{(h',w')\in\mathcal{P}_{k,i}} \gamma_i^{h'w'}} \right\|_2$$
$$\leq \frac{\sum_{(h',w')\in\mathcal{P}_{k,i}} \gamma_i^{h'w'} \|h_i^{h'w'}\|_2}{\sum_{(h',w')\in\mathcal{P}_{k,i}} \gamma_i^{h'w'}} \tag{38}$$
$$\leq R$$

Therefore:

$$\|h_i^{hw} - c_{k,i}\|_2 \leq 2R \tag{39}$$

This gives us the upper bound:

$$\text{DCD}_i^{hw} \leq \frac{(2R)^2}{\epsilon} = \frac{4R^2}{\epsilon} \tag{40}$$

For gradient stability, consider the derivative with respect to $h_i^{hw}$:

$$
\begin{aligned}
\nabla_{h_i^{hw}} \text{DCD}_i^{hw} = & \frac{1}{\|h_i^{hw} - c_{\text{bd},i}\|_2 + \epsilon} \times \\
& \left[ \frac{h_i^{hw} - c_{\text{fg},i}}{\|h_i^{hw} - c_{\text{fg},i}\|_2} \cdot \|h_i^{hw} - c_{\text{bg},i}\|_2 \right. \\
& \left. + \frac{h_i^{hw} - c_{\text{bg},i}}{\|h_i^{hw} - c_{\text{bg},i}\|_2} \cdot \|h_i^{hw} - c_{\text{fg},i}\|_2 \right] \\
& - \frac{\text{DCD}_i^{hw} \cdot (h_i^{hw} - c_{\text{bd},i})}{(\|h_i^{hw} - c_{\text{bd},i}\|_2 + \epsilon)^2}
\end{aligned}
\tag{41}
$$

Each term is bounded by construction, ensuring:

$$\|\nabla_{h_i^{hw}} \text{DCD}_i^{hw}\|_2 \leq \frac{6R}{\epsilon} \tag{42}$$

This bound guarantees stable gradient flow through the DCD mechanism.

### A.4.3 JOINT OPTIMIZATION CONVERGENCE

**Theorem 3.** The unified objective combining meta-learning and boundary refinement converges to a stationary point under the combined action of both mechanisms.

**Proof.** The total gradient with respect to model parameters is:

$$
\begin{aligned}
\nabla_\theta \mathcal{L}_{\text{total}} = & \sum_{i=1}^{N} \sum_{h,w} \nabla_\theta \mathcal{L}_i^{hw}(\theta) \\
& + \lambda_1 \sum_{i=1}^{N} \sum_{(h,w) \in \mathcal{P}_{\text{bd},i}} \left[ w_i^{hw} \nabla_\theta \mathcal{L}_i^{hw}(\theta) \right. \\
& \left. + \mathcal{L}_i^{hw}(\theta) \nabla_\theta w_i^{hw} \right] \\
& + \lambda_2 \nabla_\theta \mathcal{L}_{\text{Dice}}(M_i, \hat{M}_i)
\end{aligned}
\tag{43}
$$

The boundary weight gradient term introduces second-order effects:

$$
\begin{aligned}
\nabla_\theta w_i^{hw} = & \frac{w_i^{hw}}{\tau_{\text{dcd}}} \left[ \nabla_\theta \text{DCD}_i^{hw} \right. \\
& \left. - \sum_{(h',w') \in \mathcal{P}_{\text{bd},i}} w_i^{h'w'} \nabla_\theta \text{DCD}_i^{h'w'} \right]
\end{aligned}
\tag{44}
$$

Given the bounded gradients established in Theorem 2 and the Lipschitz continuity of the loss functions, the composite gradient satisfies:

$$\|\nabla_\theta \mathcal{L}_{\text{total}}\|_2 \leq C_1 + \lambda_1 C_2 + \lambda_2 C_3 \tag{45}$$

where $C_1, C_2, C_3$ are constants dependent on the problem parameters. This boundedness, combined with the monotonic decrease property from Theorem 1, ensures convergence to a stationary point.

The theoretical analysis confirms that MetaDCSeg achieves robust segmentation by maintaining stable optimization dynamics while adaptively handling both label noise and boundary uncertainty.

## A.5 Numerical Stability and Complexity Analysis

### A.5.1 Numerical Stability of DCD Computation

The Dynamic Center Distance (DCD) metric involves division by the distance to the boundary center, which raises potential numerical stability concerns. We analyze three critical scenarios that may arise during computation and propose corresponding solutions.

**Case 1: Near-boundary pixels.** When $\mathbf{h}_i^{hw} \approx \mathbf{c}_{\text{bd},i}$, the denominator $\|\mathbf{h}_i^{hw} - \mathbf{c}_{\text{bd},i}\|_2$ approaches zero. Although the stability constant $\epsilon = 10^{-8}$ prevents division by zero, it may still lead to extremely large DCD values that destabilize training. To address this issue, we implement a clipping mechanism:

$$\text{DCD}_i^{hw} = \min\left( \frac{\|\mathbf{h}_i^{hw} - \mathbf{c}_{\text{fg},i}\|_2 \cdot \|\mathbf{h}_i^{hw} - \mathbf{c}_{\text{bg},i}\|_2}{\|\mathbf{h}_i^{hw} - \mathbf{c}_{\text{bd},i}\|_2 + \epsilon}, \text{DCD}_{\max} \right), \tag{46}$$

where $\text{DCD}_{\max} = 100$ is empirically determined to maintain numerical stability while preserving discriminative power.

**Case 2: Degenerate center configurations.** Another critical scenario occurs when any region $\mathcal{P}_{k,i}$ contains very few pixels or all pixels have low reliability weights $\gamma_i^{hw}$, causing the center become unreliable. We handle this situation through a fallback mechanism:

$$\mathbf{c}_{k,i} = \begin{cases} \frac{\sum_{(h,w) \in \mathcal{P}_{k,i}} \gamma_i^{hw} \cdot \mathbf{h}_i^{hw}}{\sum_{(h,w) \in \mathcal{P}_{k,i}} \gamma_i^{hw}}, & \text{if } |\mathcal{P}_{k,i}| \geq \tau_{\min} \\ \mathbf{c}_{k,\text{default}}, & \text{otherwise} \end{cases} \tag{47}$$

where $\tau_{\min} = 10$ pixels and $\mathbf{c}_{k,\text{default}}$ are pre-computed default centers from the training set statistics.

**Case 3: Gradient flow.** The softmax normalization with temperature $\tau_{\text{dcd}}$ ensures bounded gradients throughout training. As $\tau_{\text{dcd}} \to 0$, the distribution becomes sharper but may cause gradient vanishing for low-DCD pixels. Therefore, we empirically set $\tau_{\text{dcd}} = 1.0$ to achieve an optimal balance between distribution sharpness and stable gradient flow.

### A.5.2 Computational Complexity Analysis

To evaluate the practical feasibility of MetaDCSeg, we analyze its computational complexity in terms of both time and space requirements.

**Time Complexity:** The meta-learning update at each iteration requires $O(|\mathcal{B}| \cdot H \cdot W \cdot C)$ for the forward pass and $O(|\mathcal{B}| \cdot H \cdot W \cdot C + |\mathcal{B}| \cdot P)$ for the backward pass, where $P$ represents the number of model parameters. Additionally, the weight update mechanism necessitates an extra validation forward pass with complexity $O(|\mathcal{V}| \cdot H \cdot W \cdot C)$. For the DCD computation, feature extraction requires $O(N \cdot H \cdot W \cdot D)$ per epoch, center calculation needs $O(N \cdot H \cdot W)$ (computed in a single pass for all three regions), and distance computation demands $O(N \cdot |\mathcal{P}_{\text{bd}}| \cdot D)$ where $|\mathcal{P}_{\text{bd}}|$ denotes the number of boundary pixels.

Combining these components, the total time complexity per epoch becomes:

$$O(N \cdot H \cdot W \cdot (C + D) + N \cdot P/|\mathcal{B}|) + O(M \cdot H \cdot W \cdot C) \tag{48}$$

Given that $M \ll N$ in our meta-learning setup and typically $|\mathcal{P}_{\text{bd}}| \approx 0.1 \cdot H \cdot W$ for medical images, the computational overhead compared to standard training is approximately 1.3× to 1.5×, which remains practical for most applications.

**Space Complexity:** The primary memory requirements consist of $O(N \cdot H \cdot W)$ for storing pixel-wise weights $\alpha_i^{hw}$ and $\beta_i^{hw}$, and $O(N \cdot H \cdot W \cdot D)$ for feature maps if cached. The storage for centers is only $O(3 \cdot D \cdot N)$, which is negligible compared to the other terms. The dominant overhead arises from pixel-wise weight storage, amounting to approximately $2 \cdot N \cdot H \cdot W \cdot \text{sizeof(float)}$ bytes, which remains manageable for typical medical imaging datasets.

