# OpenReview forum: "MetaDCSeg: Robust Medical Image Segmentation via Meta Dynamic Center Weighting"
_ICLR.cc/2026/Conference — ICLR 2026 Conference Withdrawn Submission_

### Official Review · Reviewer_zpTC · 2025-10-20

**Soundness:** 2
**Presentation:** 2
**Contribution:** 2
**Rating:** 4
**Confidence:** 5

**Summary:**

The paper proposes MetaDCSeg, a synergistic two-stage approach for robust medical image segmentation under label noise and boundary ambiguity. It introduces meta dynamic center weighting to address spatially heterogeneous label noise. Experiments are reported on four datasets: MSD Heart, MSD Brain, PROMISE12, and Kvasir-SEG.

**Strengths:**

- The problem of robust segmentation under label noise and boundary ambiguity is clinically relevant and well-motivated in medical imaging.
  - The proposed MetaDCSeg framework integrates dynamic thresholding with meta-learning-inspired center weighting, offering a potentially novel angle on handling spatially varying label reliability.
  - Evaluation across four diverse medical segmentation datasets (MSD Heart/Brain, PROMISE12, Kvasir-SEG) demonstrates broad applicability.
  - The inclusion of FLOPs as an efficiency metric shows awareness of computational practicality, though its calculation method is unclear.

**Weaknesses:**

- The definition of “noise” conflates subjective annotation errors (e.g., inconsistent boundaries) with objective imaging artifacts (e.g., device heterogeneity, lighting changes); the manuscript does not justify this unification theoretically or empirically. (Introduction, second paragraph)
  - Critical methodological components lack explanation: Fig. 1 is never referenced in the text; Fig. 2 uses γ_{h,w} with subscript notation inconsistent with α^t and β^t in the main text, and its “meta-learning” block has no corresponding description. (Fig. 1, Fig. 2)
  - Reproducibility is compromised: the anonymous GitHub lacks a README and pretrained weights.
  - Key implementation details are missing: how D_val is constructed, how pseudo-labels y_i^{pseudo,hw} are computed, and how the noise ratios for each dataset are quantified.
  - The motivation for choosing meta dynamic center weighting over alternatives like uncertainty learning or prototype learning is not discussed.
  - The relationship between the proposed method and prior work such as Lei et al. (2025) is unclear. Specifically, why class-imbalance-aware dynamic thresholds are insufficient.

**Questions:**

- What is the precise scope of “noise” addressed by MetaDCSeg? Does it include imaging artifacts (e.g., scanner-induced artifacts, lighting variation), or is it limited to label-level inconsistencies such as boundary ambiguity and annotation errors? Please clarify the unifying assumption that permits treating these heterogeneous sources as a single “noise” problem.
  - How are the noise ratios for MSD Heart, MSD Brain, PROMISE12, and Kvasir-SEG computed? What metric or procedure quantifies the degree of label noise or boundary ambiguity in these datasets?
  - How is the validation set D_val obtained? Is it a held-out clean subset? If so, how is its cleanliness guaranteed under the assumption of pervasive label noise?
  - How is y_i^{pseudo,hw} generated? Is it derived from model predictions, external models, or heuristic rules?
  - What is the computational overhead of the online meta-weighting algorithm? How does the performance of MetaDCSeg compare to a variant with fixed (non-meta) weights?
  - Why is meta dynamic center weighting preferred over other noise-robust paradigms (e.g., uncertainty learning, co-teaching, prototype calibration)? A brief discussion of design alternatives would strengthen the methodological justification.
  - Please reconcile the notation discrepancy: in the text, α and β use superscripts (e.g., α^t), but in Fig. 2, γ uses subscript (γ_{h,w}). Also, what role does the “meta-learning” module in Fig. 2 play, and where is it described in the main text?

---

### Official Review · Reviewer_4RvA · 2025-10-27

**Soundness:** 2
**Presentation:** 2
**Contribution:** 2
**Rating:** 2
**Confidence:** 4

**Summary:**

The paper proposes MetaDCSeg, a framework for dynamically learning pixel-wise weights intended to suppress prevalent label noise in medical image segmentation. The paper does this by focusing the training objective on hard-to-segment pixwels near ambiguous boundaries. The authors evaluate this on four datasets, and yield the best performance with respect to their baselines.

**Strengths:**

The paper deals with an important problem in trying to address label ambiguity, which is prevalent in medical image segmentation.

**Weaknesses:**

The evaluation of the paper is insufficient to prove the effectiveness of their method. This is largely due to the space of medical image segmentation being very mature and having various strong segmentation baselines.
In particular, the nnU-Net method [1] has established itself as a strong staple in the field, automatically configures itself to the dataset. It's widely adopted and a trusted baseline that should be include as it provides a reliable point of reference for the remaining values.
Additionally, the amount of datasets with 4 datasets is far from being comprehensive enough to allow drawing reliable conclusions. In particular, as many datasets are small (Medical Segmentation Decathlon (MSD) Heart and  Promise12) or noisy (MSD Brain). Moreover, the authors did not conduct a 5-fold cross-validation which is common, which would reduce the potential randomness of their results.

Additionally absolute values reached of the proposed method for MSD Heart are lower than the original values reported by nnU-Net [1] in their supplement which reaches 93.28 Dice, and the values reached for MSD Brain are substantially higher than the values of nnU-Net ~74 Dice while the authors reach 87.55 Dice. I am not sure where this discrepancy originates from, but it's likely a data-splitting or the authors just evaluate a subset of classes.

**Recommendations:**
The burden of proof in semantic segmentation for medical image segmentation is high, so to alleviate that and show this method clearly advances the state-of-the-art more reliable baseline methods should be included.
There has been some recent benchmarks that investigate this and which methods should be included [2,3] and are state of the art.
Additionally, the authors should extend the amount of datasets to more and larger pathological datasets. I.e. the BraTS challenge has had some brain metastasis datasets with challenging pathologies or there was the recent Panther challenge in this years MICCAI with pancreatic tumours.

[1]: Isensee, F., Jaeger, P.F., Kohl, S.A.A. et al. nnU-Net: a self-configuring method for deep learning-based biomedical image segmentation. Nat Methods 18, 203–211 (2021). https://doi.org/10.1038/s41592-020-01008-z

[2]: Isensee, Fabian, et al. "nnu-net revisited: A call for rigorous validation in 3d medical image segmentation." International Conference on Medical Image Computing and Computer-Assisted Intervention. Cham: Springer Nature Switzerland, 2024.

[3]: Bassi, P. R., Li, W., Tang, Y., Isensee, F., Wang, Z., Chen, J., ... & Zhou, Z. (2024). Touchstone benchmark: Are we on the right way for evaluating ai algorithms for medical segmentation?. Advances in Neural Information Processing Systems, 37, 15184-15201.

**Questions:**

The current method evaluates 2D and 3D datasets.
Were the 3D datasets trained with an architecture that was adapted to the 3D or were the 3D datasets trained in a slice-wise fashion?

---

### Official Review · Reviewer_qtW3 · 2025-10-31

**Soundness:** 3
**Presentation:** 2
**Contribution:** 2
**Rating:** 4
**Confidence:** 4

**Summary:**

(1) Propose MetaDCSeg, a novel framework that seamlessly integrates pixel-wise metalearning with dynamic center weighting to address both label noise and boundary ambiguity in medical image segmentation, enabling robust learning under noisy supervision

(2) Introduce a Dynamic Center Distance (DCD) mechanism that quantifies boundary uncertainty through weighted feature distances to foreground, background, and boundary centers, combined with a meta-learning paradigm that dynamically learns pixel-wise weights

**Strengths:**

(1) The method is well-described, and equations and figures help readers understand authors' idea.

(2) Authors evaluated their methods in four datasets with different modalities to demonstrate the effectiveness of their methods.

**Weaknesses:**

(1) The current evaluation is not sufficient. Authors reported the mean DSC scores, but it is necessary to report the standard deviations to show whether their methods is robust and whether it will demonstrate large variations to noisy labels. Additionally, the MSD heart is a small dataset, and it only includes 20 CT scans. Thus, it is necessary to evaluate their methods on other CT datasets with a larger size. Multi-organ segmentation evaluation is more popular and challenging than single organ segmentation. Thus, it is necessary to demonstrate the results in abdominal multi-organ segmentation datasets.

(2) Authors need to evaluate the generalizability of their method by incorporating it other architecture, such as vision transformer or hybrid CNN-ViT architectures.

**Questions:**

(1) The current evaluation is not sufficient. Authors reported the mean DSC scores, but it is necessary to report the standard deviations to show whether their methods is robust and whether it will demonstrate large variations to noisy labels. Additionally, the MSD heart is a small dataset, and it only includes 20 CT scans. Thus, it is necessary to evaluate their methods on other CT datasets with a larger size. Multi-organ segmentation evaluation is more popular and challenging than single organ segmentation. Thus, it is necessary to demonstrate the results in abdominal multi-organ segmentation datasets.

(2) Authors need to evaluate the generalizability of their method by incorporating it other architecture, such as vision transformer or hybrid CNN-ViT architectures.

---

### Official Review · Reviewer_bqC4 · 2025-11-01

**Soundness:** 3
**Presentation:** 3
**Contribution:** 2
**Rating:** 4
**Confidence:** 4

**Summary:**

The  paper study robust medical image segmentation under noisy annotations and ambiguous boundaries. The authors propose a framework called MetaDCSeg, which combines meta-learning for noise-robust segmentation with a boundary-handling mechanism. In a two-stage approach, the method first uses a pixel-wise meta-learning paradigm to reweight training losses, suppressing the influence of noisy labels by learning optimal pixel weights via a small clean validation set. Second, it introduces a Dynamic Center Distance (DCD) module to explicitly model boundary uncertainty: the network computes weighted feature distances to foreground, background, and boundary “centers,” directing attention to hard-to-segment boundary pixels. Experiments on four medical segmentation benchmarks (heart MRI, brain MRI, colonoscopy, prostate MRI) with simulated noise (20%, 40%, 60% label corruption) demonstrate improved performance. The proposed method consistently outperforms several state-of-the-art robust segmentation methods (e.g. L2B, ConDSeg, HAMIL, etc.), achieving higher mIoU and Dice scores.

**Strengths:**

1. The paper studies an important and practical problem of robust medical image segmentation under noisy annotations. The proposed approach is reasonable and well-motivated, and the specific focus on boundary regions is particularly critical given the ambiguity often present in medical segmentation tasks.

2. The experiments are comprehensive, with comparisons against a diverse set of baseline methods under varying noise levels. In addition, the ablation studies on different components provide valuable insight into the contribution and effectiveness of each module.

3. Overall, the paper is well-written, with clear exposition and high presentation quality. The methodology is articulated in a structured manner, making it easy to follow the technical contributions and experimental setup.

**Weaknesses:**

1. The novelty of the method is somewhat limited, as meta-learning has already been employed in prior work to address learning from noisy labels [a1]. In addition, earlier studies such as [a2] have also explored strategies for handling boundary ambiguity in the context of noisy semantic segmentation. However, the paper lacks discussion and empirical comparison with these relevant approaches, which weakens the positioning of its contributions.

2. The method requires a small but clean validation subset for meta-optimization, which may not always be available in medical datasets.

3. Experiments use U-Net++ as the main backbone; robustness on transformer-based or large foundation models remains unexplored.

4. Performance depends on multiple hyperparameters (e.g., threshold, temperature) that may require tuning per dataset.

[a1] Ren, Mengye, et al. "Learning to reweight examples for robust deep learning." ICML, 2018.
[a2] Li, Shuailin, et al. "Superpixel-guided iterative learning from noisy labels for medical image segmentation." MICCAI, 2021.

**Questions:**

Please refer to the weaknesses section.

---

### Note · Authors · 2026-01-15

I have read and agree with the venue's withdrawal policy on behalf of myself and my co-authors.